# GPCR voltage dependence controls neuronal plasticity and behavior

Eyal Rozenfeld[1,2], Merav Tauber[3], Yair Ben-Chaim[3] & Moshe Parnas [1,2✉]

G-protein coupled receptors (GPCRs) play a paramount role in diverse brain functions. Almost 20 years ago, GPCR activity was shown to be regulated by membrane potential in vitro, but whether the voltage dependence of GPCRs contributes to neuronal coding and behavioral output under physiological conditions in vivo has never been demonstrated. Here we show that muscarinic GPCR mediated neuronal potentiation in vivo is voltage dependent. This voltage dependent potentiation is abolished in mutant animals expressing a voltage independent receptor. Depolarization alone, without a muscarinic agonist, results in a nicotinic ionotropic receptor potentiation that is mediated by muscarinic receptor voltage dependency. Finally, muscarinic receptor voltage independence causes a strong behavioral effect of increased odor habituation. Together, this study identifies a physiological role for the voltage dependency of GPCRs by demonstrating crucial involvement of GPCR voltage dependence in neuronal plasticity and behavior. Thus, this study suggests that GPCR voltage dependency plays a role in many diverse neuronal functions including learning and memory.

[1] Department of Physiology and Pharmacology, Sackler School of Medicine, Tel Aviv University, Tel Aviv 69978, Israel. [2] Sagol School of Neuroscience, Tel Aviv University, Tel Aviv 69978, Israel. [3] Department of Natural and Life Sciences, The Open University of Israel, Ra'anana 43107, Israel. ✉email: mparnas@tauex.tau.ac.il

G protein coupled receptors (GPCRs) comprise the largest known family of transmembrane receptors, and are present in all life forms[1]. Over 90% of non-sensory GPCRs are expressed in the brain, where they mediate responses to various biologically active molecules including acetylcholine, glutamate, dopamine, noradrenaline, serotonin, histamine, GABA, peptides, lipid-derived products, and also to mechanical stimuli[2,3]. As such, they play a paramount role in diverse brain functions, for example, vision, taste, olfaction, behavior regulation, neuromodulation, and regulation of the immune system among others. GPCRs thus mediate a large variety of brain and body functions and are critical for normal brain function[4–6].

GPCRs are activated by binding of specific agonists. They then interact with GTP-binding proteins (G-proteins) and mediate their effect via a range of second messenger signaling pathways in the cell[4–6]. Almost 20 years ago, the activity level, as well as agonist binding affinity, of several GPCRs were shown to be regulated by membrane potential in vitro[7–15]. For example, the activity of cholinergic $M_2$ muscarinic receptors ($M_2R$)[10], metabotropic glutamate receptors[14], and α2 adrenergic receptors[12] are reduced by depolarization. In contrast, the activity of $M_1$ muscarinic receptors ($M_1R$)[10] and mGluR1a glutamate receptors[14] are increased by depolarization. GPCR voltage dependency is about an order of magnitude weaker than that of voltage-gated channels[16], and GPCRs lack the type of voltage sensor composed of positively charged residues, seen in voltage-gated channels[11].

Despite several decades of research into the functions of GPCRs, it is still not clear whether GPCR voltage dependency plays a role in vivo under physiological conditions or contributes to neuronal coding and behavioral output. Although GPRC voltage dependence was shown to control synaptic release initiation and duration in vitro[17–22], there is no evidence that these small changes in the duration of synaptic release affect neuronal computation or behavioral output, especially in the background of noisy neural activity. Thus, the question of whether GPCR voltage dependency plays a role in vivo under physiological conditions and contributes to neuronal coding and behavioral output remains open.

Currently, the only known voltage sensor of GPCRs is for muscarinic receptors. The Drosophila olfactory system is cholinergic and has high expression levels of muscarinic receptors[23]. Only two Drosophila muscarinic receptors are expressed in the fly brain: a $G_q$ coupled Drosophila muscarinic type A receptor (mAChR-A, homologous to $M_1R$); and a $G_{i/o}$ coupled Drosophila muscarinic type B receptor (mAChR-B, homologous to $M_2R$)[24,25]. Although the Drosophila genome contains a third Drosophila muscarinic type C receptor, its expression level in the brain is negligible[26,27]. As a result, manipulation of Drosophila muscarinic receptors results in profound physiological and behavioral effects[23,28], which makes, the Drosophila olfactory system well suited to examine whether GPCR voltage dependence affects physiological processes and behavior.

Odors activate Drosophila cholinergic olfactory receptor neurons (ORNs) that are located in the antennae and maxillary palps. Each ORN expresses a single odorant receptor gene[29,30]. ORNs expressing the same receptor send their axons to a single glomerulus in the antennal lobe (AL, homologous to the mammalian olfactory bulb)[31–33]. Second-order excitatory cholinergic projection neurons (ePNs) send their dendrites to a single glomerulus and serve as the primary output channel of the AL[34]. The AL also contains multi glomeruli inhibitory GABAergic and glutamatergic local neurons (iLNs)[35,36] that receive input from ORNs and PNs.

We recently demonstrated that mAChR-A is mainly localized to a subpopulation of GABAergic iLNs where it induces short-term potentiation of iLNs[23]. Knockdown of mAChR-A in iLNs

also causes changes of odor valence[23]. In addition to odor valence, iLN activity is required for odor habituation, sustained odor responses, regulating gain control, signal separation, and enhancement of interglomerular contrast[36–43].

Here we show that the Drosophila mAChR-A, which diverged from the mammalian muscarinic receptors over 700 million years ago[24] is also voltage dependent. We use Clustered Regularly Interspaced Short Palindromic Repeats (CRISPR/Cas9) to generate a fly strain with two point mutations that abolish the mAChR-A voltage dependence while retaining normal maximal activity. This enables us to demonstrate that wt mAChR-A induced post-tetanic potentiation (PTP) is voltage dependent and that this dependency is abolished in mutant flies expressing a voltage independent mAChR-A. Even more striking is that depolarization alone, without any agonist, causes a potentiation of nicotinic receptors in iLNs which is mediated by mAChR-A. In addition, the generation of a voltage-independent mAChR-A results in a pronounced behavioral effect of increased odor habituation, which was localized to iLNs. Taken together, this study provides a demonstration of a physiological role for the voltage dependency of GPCRs, and may serve as a paradigm shift in our understanding of neural function and drug discovery.

## Results

**The Drosophila mAChR-A GPCR is voltage dependent.** The Drosophila mAChR-A has overall low sequence homology to its mammalian homolog $M_1R$[24]. We therefore first examined the voltage dependency of mAChR-A expressed in Xenopus oocytes. As for $M_1R$, depolarization (+40 mV) increased mAChR-A activity as indicated by a leftward shift in the mAChR-A dose–response curve compared to the curve obtained by hyperpolarization (−80 mV, Fig. 1A). Thus, when the membrane potential is depolarized, mAChR-A is in a high activity state, and when the membrane potential is hyperpolarized, it is in a low activity state. In order to isolate the effects of the voltage dependency of mAChR-A from other voltage-related effects, we sought to generate a voltage-independent mAChR-A. Direct gating current measurements revealed that the voltage sensor of mammalian muscarinic receptors consists of three tyrosine residues (Supplementary Fig. 1A)[44]. Replacing these residues with the electrically neutral amino acid, alanine, abolished both the gating currents and the voltage-dependent activity of $M_2R$[44]. Attempts at replacing these residues in mAChR-A resulted in impaired receptor activity unsuitable for further experiments (Supplementary Fig. 1).

Since these attempts to generate a voltage-independent receptor proved unsuccessful, we sought a different approach. The N- and C- terminals of the third intracellular loop of $M_1R$ and $M_2R$ are required for G-protein coupling[45,46], and replacing the $M_2R$ N-terminal residues KKDKK with the $M_1R$ N-terminal residues ELAAL was shown to abolish the $M_2R$ voltage dependence while retaining receptor function[11]. Since the homologous motif in mAChR-A is KDLPN (Supplementary Fig. 1C), we replaced aspartic acid at position 301 and proline at position 303 with lysine (D301K, P303K) to generate mAChR-A with a final motif sequence of KKLKN. This modified receptor that we term mAChR-A-KK, possesses voltage-independent activity resembling that of mAChR-A in the high activity state, i.e. when depolarized (Fig. 1B). Importantly, the response of mAChR-A-KK to saturating levels of ACh resembles that of wt mAChR-A at the two membrane potentials tested (Fig. 1D), indicating that the overall activity of mAChR-A-KK is not impaired. Similarly, measuring endogenous $Ca^{2+}$ activated $Cl^-$ channels, activated by $G_q$ protein[10] yielded the same results (Supplementary Fig. 2A, B). We also verified that receptor

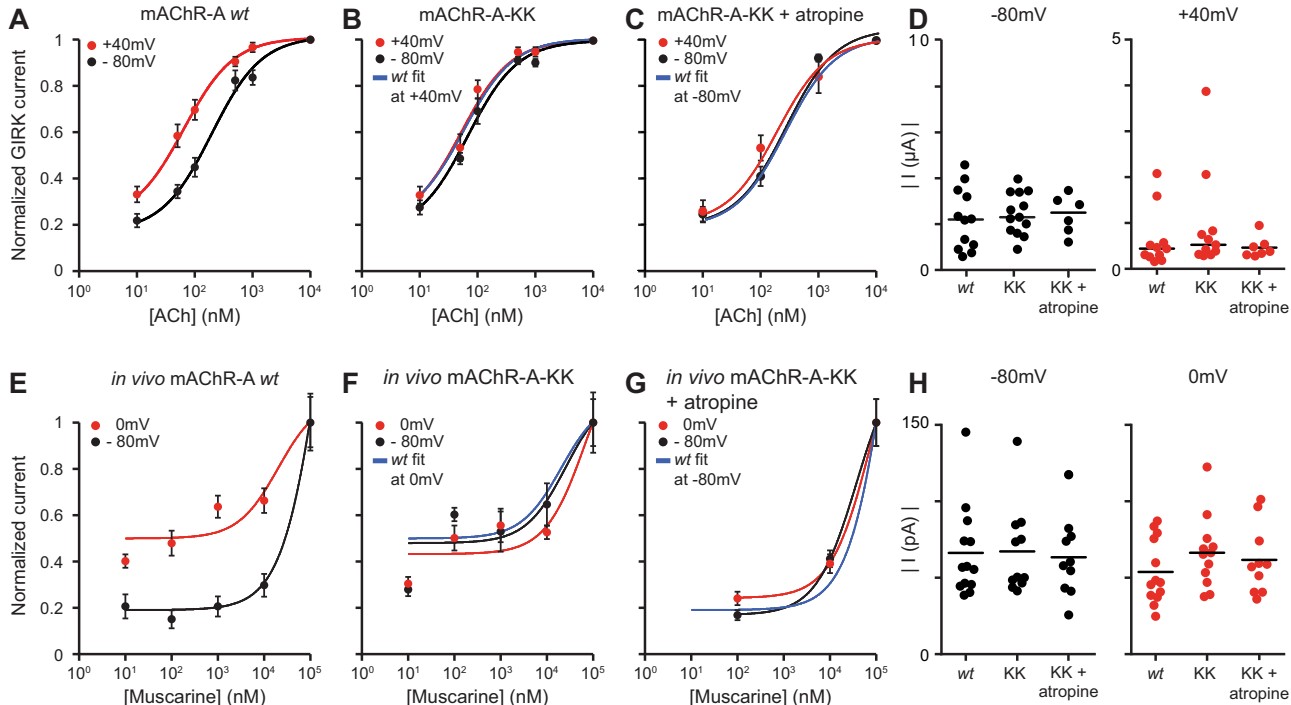

**Fig. 1 in vitro *and* in vivo voltage dependence of mAChR-A. A** Dose-response curves for *wt* mAChR-A-evoked GIRK currents in oocytes at −80 mV (black; $n = 24$) and +40 mV (red; $n = 17$). The curve of the depolarized membrane potential is shifted to the left indicating a higher activity state. **B** Dose-response curves for mAChR-A-KK-evoked GIRK currents in oocytes at −80 mV (black; $n = 23$) and +40 mV (red; $n = 19$). mAChR-A-KK is in the high activity state as indicated by the overlap with the dose–response curve of *wt* mAChR-A at +40 mV (blue line). **C** Dose–response curves for mAChR-A-KK-evoked GIRK currents in oocytes at −80 mV (black; $n = 6$) and +40 mV (red; $n = 7$) in the presence of 1 nM of atropine. Atropine shifts mAChR-A-KK to the lower activity state as indicated by the overlap with the curve of *wt* mAChR-A at −80 mV (blue line). **D** Absolute current values of *wt* mAChR-A, mAChR-A-KK, and mAChR-A-KK in the presence of 1 nM atropine, for ACh evoked currents in oocytes at −80 mV (left, black; $n = 12, 13,$ and 6, respectively) and +40 mV (right, red; $n = 11, 11$ and 7, respectively) elicited by a saturating level of ACh. Data are un-normalized values of panels **A**–**C** at $10^4$ nM ACh. **E** In vivo dose-response curves for *wt* flies of muscarine evoked currents in iLNs at −80 mV (black; $10^1$ nM $n = 11, 10^2$ nM $n = 9, 10^3$ nM $n = 9, 10^4$ nM $n = 14, 10^5$ nM $n = 13$) and 0 mV (red; $10^1$ nM $n = 9, 10^2$ nM $n = 10, 10^3$ nM $n = 10, 10^4$ nM $n = 14, 10^5$ nM $n = 13$). **F** In vivo dose–response curves for mAChR-A-KK fly strain of muscarine evoked currents in iLNs at −80 mV (black; $10^1$ nM $n = 11, 10^2$ nM $n = 13, 10^3$ nM $n = 11, 10^4$ nM $n = 11, 10^5$ nM $n = 11$) and 0 mV (red; $10^1$ nM $n = 11, 10^2$ nM $n = 18, 10^3$ nM $n = 18, 10^4$ nM $n = 12, 10^5$ nM $n = 12$). **G** In vivo dose–response curves for mAChR-A-KK fly strain of muscarine evoked currents in iLNs at −80 mV (black; $10^2$ nM $n = 11, 10^4$ nM $n = 12, 10^5$ nM $n = 10$) and 0 mV (red; $10^2$ nM $n = 12, 10^4$ nM $n = 11, 10^5$ nM $n = 10$) in the presence of 1 µM atropine. Atropine shifts mAChR-A-KK to the lower activity state as indicated by the overlap with the dose–response curve of *wt* mAChR-A at −80 mV (blue line). **H** Absolute current values of in vivo *wt* (−80mV, $n = 13$, 0 mV, $n = 13$), mAChR-A-KK (−80mV, $n = 11$, 0 mV, $n = 12$), and mAChR-A-KK flies in the presence of 1 µM atropine (−80mV, $n = 10$, 0 mV, $n = 10$), for muscarine evoked currents in iLNs at −80 mV (left, black) and 0 mV (right, red) elicited by a saturating level of muscarine. Each dot represents a single fly. Data are un-normalized values of panels **E**–**G** at $10^5$ nM muscarine. For all panels, error bars represent the standard error of the mean (SEM). Source data are provided as a Source Data file.

recruitment of the downstream $G_q$ pathway was not impaired (Supplementary Fig. 2B, C) and that receptor desensitization by β-arrestin and G protein-coupled receptor kinase 3 (GRK3) co-expressed in the oocyte was not affected in mAChR-A-KK (Supplementary Fig. 3). In order to obtain the voltage-independent mAChR-A-KK in the low activity state, we applied a low concentration of the competitive antagonist, atropine. In contrast to the situation in mammalian muscarinic receptors, atropine is specific for mAChR-A in flies and does not affect mAChR-B, allowing us to exploit this method also in vivo[25]. Application of atropine shifted the mAChR-A-KK dose–response curve to the right, so that it overlapped with the activity of *wt* mAChR-A in the low activity state (Fig. 1C). Importantly, even in the presence of atropine, the response of mAChR-A-KK to saturating levels of acetylcholine (ACh) was not different from that of the *wt* mAChR-A at any of the tested membrane potentials (Fig. 1D). Thus, by using two complementary approaches of pharmacology and gene editing, we generated a

voltage-independent mAChR-A-KK receptor that has normal maximal activity and can reside in a high or low activity state.

Even though the voltage dependence of GPCRs has been known for years, the physiological relevance to in vivo processes remains unknown. To examine the effects of mAChR-A voltage dependence on physiological processes and behavioral output, we used CRISPR to generate a fly strain that has the two-point mutations (i.e. D301K, P303K) that produce mAChR-A-KK. We have previously shown that the muscarinic agonist muscarine causes an increase in $Ca^{2+}$ levels in iLNs[23], probably by activation of $Ca^{2+}$ channels[47], resulting in an inward current flow. We now used these muscarine-induced currents to verify that *wt* mAChR-A is voltage dependent in vivo and to examine whether the D301K, P303K point mutations generate a voltage-independent mAChR-A in vivo. Similar to the results obtained with oocytes, in vivo mAChR-A activity was voltage dependent in *wt* flies and voltage independent in the mAChR-A-KK fly strain (Fig. 1E–G). In addition, the *wt* and mutant receptors responded in the same

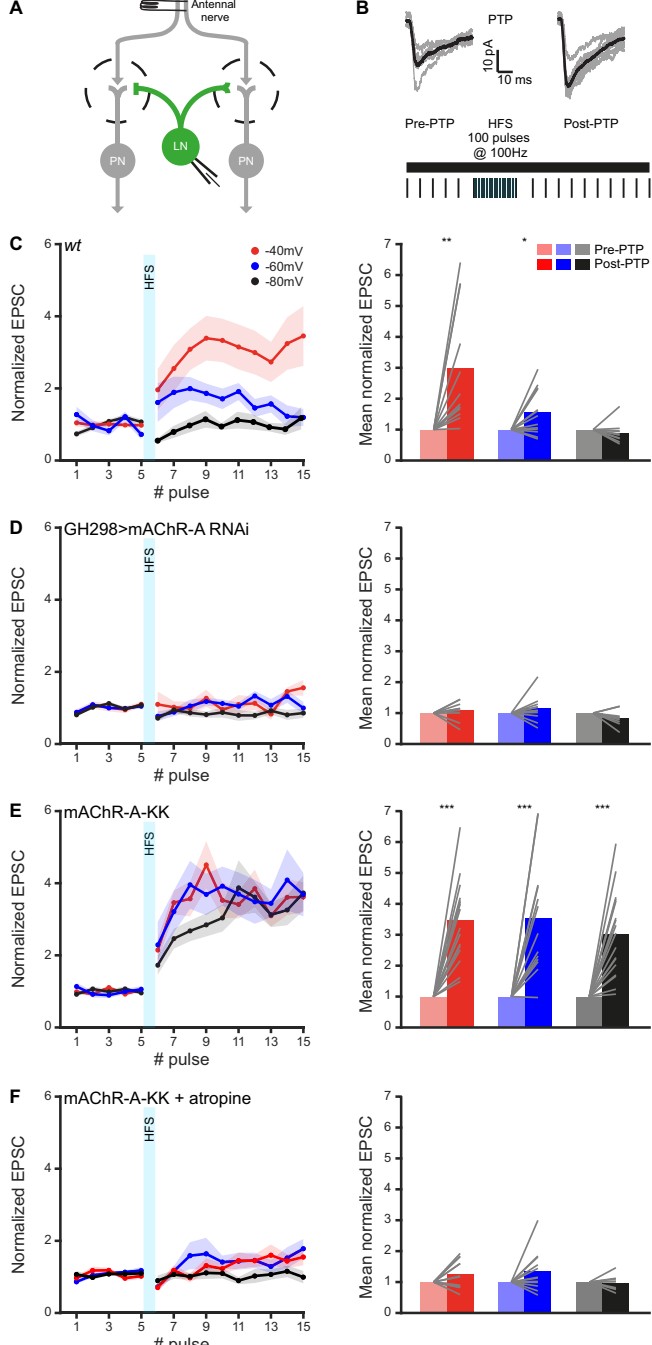

**Fig. 2 mAChR-A induced potentiation is voltage dependent. A** Experimental configuration. ORNs axons were stimulated at varying frequencies and the iLN post-synaptic evoked currents were measured using in vivo whole-cell voltage clamp. **B** Post-Tetanic Potentiation (PTP) protocol. Representative single fly traces (gray) of excitatory postsynaptic current (EPSCs) obtained before (left) and after (right) the PTP protocol. The mean EPSC is shown in black. **C** *Left*, normalized EPSC before and after PTP obtained in *wt* flies for holding potential of −80 mV (black; $n = 10$), −60 mV (blue; $n = 13$), and −40 mV (red; $n = 12$). *Right*, mean normalized EPSC obtained from the data presented on the left. Depolarizing iLNs during the HFS part of the PTP protocol increased potentiation ($p(−60 \text{ mV}) = 0.016$, $p(−40 \text{ mV}) = 0.004$, One sample two-sided *t*-test). **D** *Left*, normalized EPSC before and after PTP obtained in flies in which mAChR-A was knocked down in iLNs for holding potential of −80 mV (black; $n = 9$), −60 mV (blue; $n = 11$), and −40 mV (red; $n = 9$). *Right*, mean normalized EPSC obtained from the data presented on the left. (n.s, One sample two-sided *t*-test). **E** *Left*, normalized EPSC before and after PTP obtained in the mAChR-A-KK fly strain for holding potential of −80 mV (black; $n = 17$), −60 mV (blue; $n = 13$), and −40 mV (red; $n = 16$). *Right*, mean normalized EPSC obtained from the data presented on the left. A strong potentiation was observed irrespective of the holding potential ($p(−40 \text{ mV}) = 0.000001$, $p(−60 \text{ mV}) = 0.0003$, $p(−80 \text{ mV}) = 0.00001$, One sample two-sided *t*-test). **F** *Left*, normalized EPSC before and after PTP obtained in the mAChR-A-KK fly strain for holding potential of −80 mV (black; $n = 9$), −60 mV (blue; $n = 12$), and −40 mV (red; $n = 13$) in the presence of 1 μM atropine. *Right*, mean normalized EPSC obtained from the data presented on the left. Application of atropine shifts mAChR-A-KK to the low activity state and abolishes voltage-independent potentiation (n.s, One sample two-sided *t*-test). For all panels, shaded error bands represent the SEM, lines represent single flies and GH298-GAL4 was used to drive UAS-GFP. Source data are provided as a Source Data file.

there was substantial potentiation at the more depolarized membrane potential, no potentiation was observed at the hyperpolarized membrane potential (Fig. 2C). Knocking down mAChR-A specifically in iLNs abolished the potentiation irrespective of membrane potential, confirming that PTP is controlled primarily by mAChR-A (Fig. 2D). No potentiation was observed when the iLN membrane potential was held at −40mV but without applying the potentiation protocol (Supplementary Fig. 4A). These results suggest that the change in membrane potential shifts mAChR-A from the low activity state to a higher activity state and this shift in activity is required for the PTP. We therefore hypothesized that since mAChR-A-KK is "stuck" at the high activity state, it should exhibit potentiation values similar to those achieved by depolarization. Similarly, applying atropine, which maintains the voltage-independent mAChR-A-KK in the low activity state (Fig. 1C, G), should result in a weaker potentiation than *wt* mAChR-A, or even produce no potentiation at all. In accordance with our hypothesis, applying the same protocol to flies carrying the mAChR-A-KK mutation, resulted in voltage-independent potentiation at all membrane potentials (Fig. 2E), which was not observed using a lower frequency stimulation (Supplementary Fig. 4B). In addition, atropine at a concentration that maintains the mAChR-A-KK in the low activity state (Fig. 1G), completely abolished neuronal potentiation (Fig. 2F). Taken together, these results demonstrate that mAChR-A voltage dependence is required for efficient recruitment of mAChRA and participates in physiological plasticity processes.

**A voltage-dependent mAChR-A and nicotinic receptors crosstalk.** Recently, $M_2R$ basal activity, which reflects spontaneous coupling of $M_2R$ to its cognate G-protein, was shown to also be voltage dependent[48]. In another study, activation of

way to saturating levels of muscarine at all the measured membrane potentials, indicating that mAChR-A-KK activity is not impaired in vivo (Fig. 1H).

**mAChR-A voltage dependency controls neuronal potentiation.** We have recently reported that mAChR-A contributes to short-term potentiation in iLNs labeled by the GH298-GAL4 driver line[23]. Having demonstrated the voltage dependency of mAChR-A activity, it was now interesting to examine whether the iLN potentiation is similarly voltage dependent. To this end, we used a PTP protocol of 100 antennal nerve stimulations at 100 Hz to ORNs that are presynaptic to iLNs (Fig. 2A, B). Postsynaptic iLNs were held at either −40 mV, −60 mV, or −80 mV during the PTP phase, and then returned to their resting membrane potential (−60 mV). Surprisingly, the results indicated that while

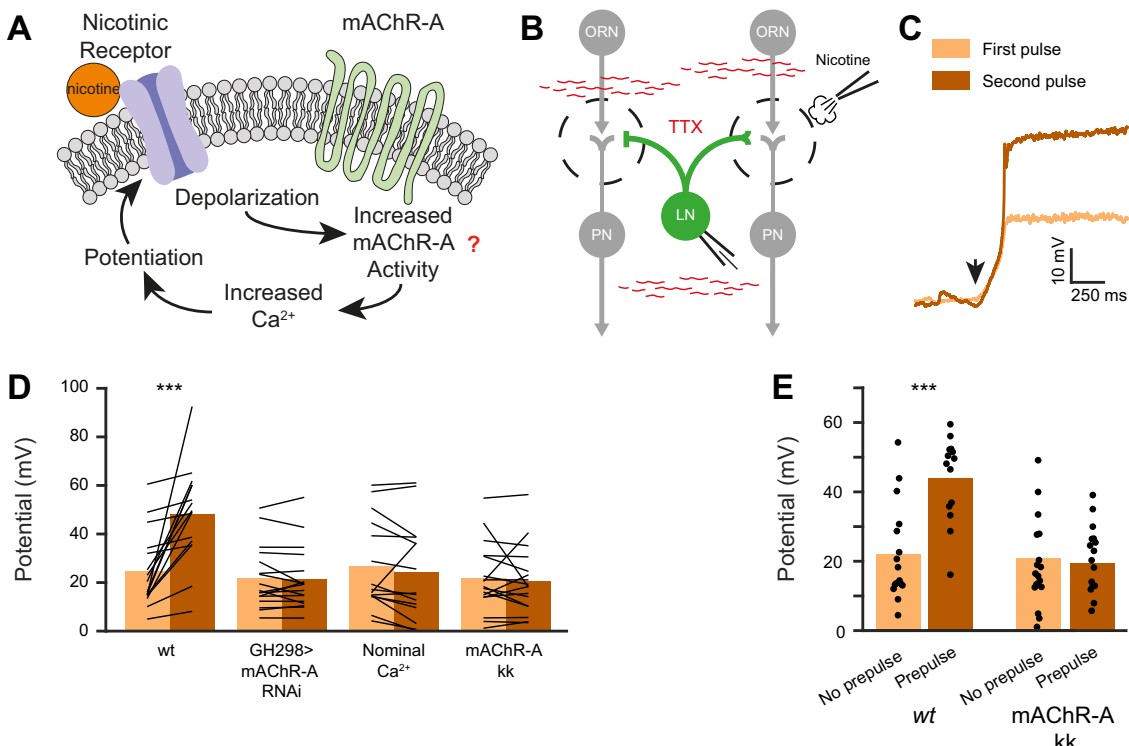

**Fig. 3 mAChR-A nicotinic induced potentiation is voltage dependent. A** A diagram of the suggested crosstalk between nicotinic receptors and mAChR-A. Depolarization induced by activation of the nicotinic receptors can increase the constitutive activity of mAChR-A. The increased mAChR-A activity results in potentiation of the nicotinic response. **B** A diagram of the experimental configuration. Network activity was blocked using TTX (1 μM), two pulses of nicotine (100 μM) were injected into the AL with a temporal interval of 1 min, and the iLN excitatory postsynaptic potential was measured using in vivo whole-cell current clamp. **C** Representative traces obtained from a single fly demonstrating the potentiation occurring in the second nicotinic pulse (dark brown) relative to the first pulse (light brown). **D** Nicotinic induced double pulse potentiation (first pulse shown in light brown, second shown in dark brown) was observed in *wt* flies ($n = 16$). No potentiation was observed when mAChR-A was knocked down in iLNs using GH298-GAL4 and UAS-mAChR-A RNAi ($n = 17$), in nominal $Ca^{2+}$ ($n = 15$), and in the mAChR-A-KK fly strain ($n = 17$). Each line represents a single fly ($p(wt) = 0.0005$, Paired sample two-sided *t*-test). **E** A depolarizing pre-pulse (40 mV for 30 s) before the nicotinic pulse potentiated the response to nicotine in *wt* flies (No pre-pulse; $n = 16$, Pre-pulse; $n = 14$) but not in the mAChR-A-KK fly strain (No pre-pulse; $n = 17$, Pre-pulse; $n = 16$). First pulse data are the same as in panel **D**. Each dot represents a single fly $p(wt) = 0.00008$, Two sample two-sided *t*-test).

mAChR-A inhibited potassium currents, thereby increasing the $Ca^{2+}$ level and eventually inducing the potentiation of the nicotinic receptors[47]. These two observations raise the possibility that depolarization induced by the activation of nicotinic receptors can increase the constitutive activity of mAChR-A and result in potentiation of the nicotinic response (Fig. 3A). We therefore examined whether two consecutive nicotine pulses would result in potentiation of the second nicotine pulse (Fig. 3B). This was indeed the case in flies expressing *wt* mAChR-A (Fig. 3C, D). This double pulse potentiation was dependent on mAChR-A activity as it was abolished when mAChR-A was knocked down (Fig. 3D). Using nominal external $Ca^{2+}$ also abolished the paired pulse potentiation, further supporting our conclusion that this potentiation involves the pathway described for mAChR-A potentiation (Fig. 3D). As there is no exogenous mAChR-A agonist, this protocol is not expected to give rise to increased potentiation in flies expressing the mAChR-A-KK mutation. Rather, since these receptors are insensitive to membrane potential, the basal activity would be expected to be unaffected by nicotine and no potentiation will be observed. Indeed, applying the paired-pulse potentiation in mAChR-A-KK flies did not elicit any potentiation to the second nicotine pulse (Fig. 3D).

These results demonstrate that activation of nicotinic receptors recruits mAChR-A which in turn potentiate the nicotinic receptors. As nicotine does not activate muscarinic receptors and since nicotine had no effect on the voltage-independent

mAChR-A-KK, the most likely explanation is that the depolarization induced by the nicotine pulse recruits mAChR-A. We therefore examined whether a depolarizing pre-pulse step prior to nicotine application would induce a similar potentiation. Indeed, a depolarizing step prior to the nicotine pulse resulted in a significant increase in the potential induced by the nicotine pulse (Fig. 3E). To verify that the potentiation following the depolarizing pre-pulse is mediated by mAChR-A, we repeated the experiment using the mAChR-A-KK fly strain. No potentiation was observed in the case of the mAChR-A-KK fly strain (Fig. 3E). We also verified that *wt* mAChR-A activity in response to muscarine, but not that of mAChR-A-KK, is potentiated following a depolarizing pre-pulse step (Supplementary Fig. 5). Together, these results indicate that the potentiation effect is mediated by mAChR-A and not by other downstream proteins (Fig. 3E). More importantly, these results elucidate a mechanism of ionotropic receptors modulation by GPCR voltage-dependent activity.

**mAChR-A voltage dependency affects odor habituation.** The cumulative results thus far demonstrate that mAChR-A voltage dependence participates in iLN potentiation, albeit, to artificial stimuli. We therefore examined whether similar results could be elicited by a physiological odor stimulus. iLN activity depresses over time[23,49,50] and long exposure to odors was suggested to potentiate the response of iLNs[41,51] (Fig. 4A). Therefore, we

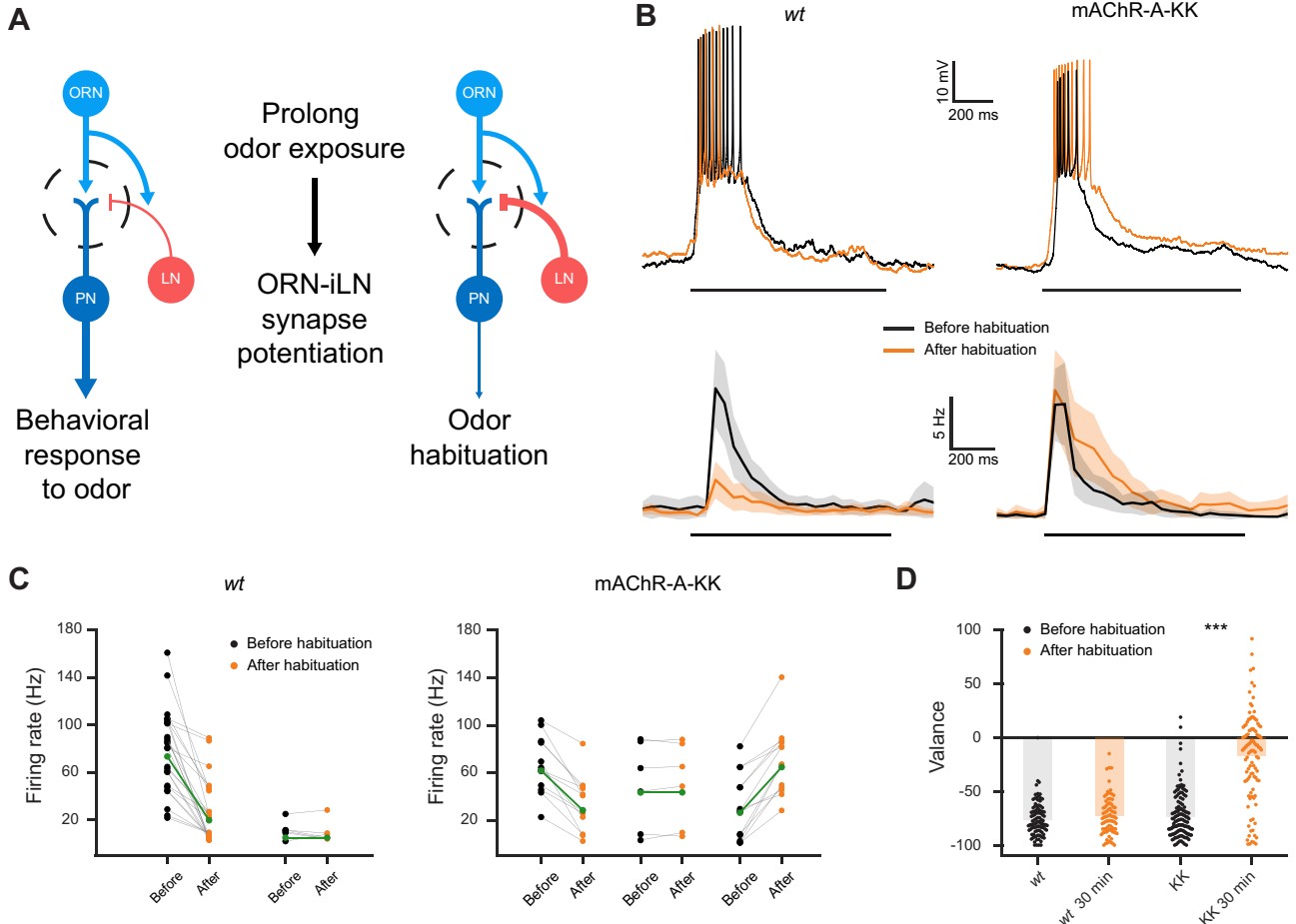

**Fig. 4 mAChR-A voltage dependence affects odor habituation. A** Proposed habituation mechanism: Before odor exposure, the ORN-iLN synapse is relatively weak. Thus, iLNs generate a modest inhibitory signal on the ORN-PN synapse, allowing PN activity. Following a long odor exposure, the ORN-iLN synapse is potentiated, blocking the transmission from ORNs to PNs, thereby decreasing PN activity, resulting in odor habituation. Line thickness indicates firing rate. **B** *Top*, Examples of the decrease in iLN firing rate observed in *wt* flies (left) and potentiation in iLN activity observed in the mAChR-A-KK fly strain (right). Black and orange denote the iLN odor response before and after odor habituation (30 min exposure) respectively. Black line indicates odor application. *Bottom*, iLN mean firing rate before and after odor habituation. Black and orange denote the iLN odor response before and after the odor habituation protocol respectively. (*wt*, $n = 31$, mAChR-A-KK, $n = 31$). While there is a strong decrease in iLN firing rate in *wt* flies ($p(wt) = 0.0000002$, Paired sample two-sided *t*-test), over all iLNs there is no difference in the mAChR-A-KK fly strain. Shaded error bands represent the SEM. **C** Maximal firing rate of the 10 stimulus trials presented to each cell, before (black) and after (orange) 30 min exposure to ethyl butyrate for data shown in panel B. Dots represent individual flies. Group averages are shown in green. In the case of *wt* flies ~ 80% of flies showed a decrease in firing rate following long odor exposure. In contrast, in the mAChR-A-KK strain, only ~ 38% of flies showed a decrease firing frequency whereas ~ 40% of flies showed potentiation. Points were randomly jittered for visualization. **D** The valence (negative values indicate aversion) flies assign to ethyl butyrate before and after odor habituation (30 min exposure). For *wt* flies, odor valance did not change, however, the mAChR-A-KK fly strain displayed a strong reduction in odor avoidance, indicating odor habituation. (*wt* before, $n = 102$, *wt* after, $n = 84$, mAChR-A-KK before, $n = 139$, mAChR-A-KK after, $n = 107$, $p < 10^{-30}$, Two sample two sided *t*-test). Source data are provided as a Source Data file.

examined the effect of 30 min exposure to the odor ethyl butyrate on iLN activity in flies expressing *wt* mAChR-A or mAChR-A-KK. The results indicated that exposure to odor significantly decreased the firing rate in ~ 80% of the iLNs examined in *wt* flies (Fig. 4B, C). In contrast, the response of iLNs from mAChR-A-KK flies was heterogeneous. Overall, when averaging all iLNs from mAChR-A-KK flies there was no difference in firing rate before and after odor exposure (Fig. 4B). However, a more detailed analysis revealed that in the case of mAChR-A-KK flies, long odor exposure decreased iLN firing rate in only ~ 38% of iLNs. Contrary to *wt* flies, there was a marked potentiation in 40% of iLNs (Fig. 4C). These results indicate potentiation of a subpopulation of iLNs in the case of the mAChR-A-KK fly strain.

These results all demonstrate that mAChR-A voltage dependence plays a role in physiological processes and in particular in

iLN potentiation following odor exposure. We therefore examined whether these changes in potentiation also affect behavioral output. Odor habituation is a decline in behavioral response following repeated or continuous exposure to an odorant[51], and has been suggested to require potentiation of iLNs[41,52–54]. Briefly, prior to a prolong odor exposure iLNs generate a weak inhibitory signal on the ORN-PN synapse, thus allowing the olfactory signal to efficiently propagate to higher brain regions. Following a long odor exposure, the ORN-iLN synapse is potentiated, resulting in a stronger iLN inhibitory signal, which efficiently blocks the transmission from ORNs to PNs. As a result, there is a decrease in PN signal to higher brain regions, resulting in odor habituation (Fig. 4A). We therefore examined whether odor habituation in mAChR-A-KK flies which have stronger iLN potentiation (Fig. 4B, C) is affected. Since mAChR-A and mAChR-A-KK

respond similarly to a saturating level of agonist (Fig. 1D, H), one would expect that if agonist levels that reach iLNs are high enough there would be no difference in odor habituation between *wt* mAChR-A and mAChR-A-KK flies. However, if as expected, the agonist levels arriving to mAChR-A on the iLNs are within the dynamic range of the dose–response curve, then one would expect to see a stronger habituation in flies with mAChR-A-KK than in *wt* mAChR-A flies. To test odor habituation we used custom linear chambers, each housing a single fly (Supplementary Fig. 6A). These chambers allow the presentation of an odor from either sides of the chamber while presenting odorless air flow from the other side of the chamber. The Air and odor streams converge at a central choice zone (see methods for a detailed description). Thus, each fly can choose whether to spend time in the area containing an odor. To calculate ethyl butyrate valance the odor was presented on alternating sides of the chamber for two minutes and the difference between the preference to the odor-containing side relative to the air-containing side was calculated (Supplementary Fig. 6B, see "Methods" for a detailed explanation). Following an initial odor valance measurement, flies were exposed to ethyl butyrate for 30 min on both sides of the chamber simultaneously and then odor valence was measured again (Supplementary Fig. 6B). The results presented in Fig. 4D and Supplementary Fig. 7A, indeed demonstrate significantly stronger habituation in mAChR-A-KK flies than in those with *wt* mAChR-A. Since a 30-minute odor exposure did not result in strong odor habituation in *wt* flies, we verified that habituation indeed occurs when the habituation period is prolonged to 1 h (Supplementary Fig. 7B). In addition, to control for a possible change in odor valance unconnected to the odor exposure, we repeated the habituation protocol without adding odor to the carrier airflow. This did not affect the odor valence in mAChR-A-KK flies (Supplementary Fig. 7C).

We showed that iLNs undergo stronger potentiation in the mAChR-A-KK strain following odor exposure (Fig. 4B, C) and that, as one would expect from a stronger iLN potentiation, odor habituation is stronger in the mAChR-A-KK strain. However, mAChR-A is broadly expressed in the brain[28]. Thus, it is possible that the effect mAChR-A-KK has on odor habituation does not arise solely from changes in iLN activity. We have previously shown that in the olfactory pathway, mAChR-A is mainly expressed and has a functional effect in iLNs of the AL and in a subset of third-order Kenyon cells (KCs) of the mushroom body (MB, Fig. 5A)[23,28] which were also shown to be involved in olfactory habituation[55]. Therefore, we performed rescue experiments in these two cell types. To this end we used the GH298-GAL4 driver which covers iLNs, and the MB247-GAL driver which covers the KC subtypes that express mAChR-A[23,28]. To rescue the effects of mAChR-A-KK we used two strategies: i. we overexpressed a *wt* mAChR-A using UAS-mAChR-A. ii. We knocked down mAChR-A-KK levels using UAS-mAChR-A RNAi[23,28]. Overexpression of a *wt* mAChR-A in iLNs should abolish the increased habituation effect in mAChR-A-KK flies since the overexpression of a voltage-dependent mAChR-A has a dominant effect. Similarly, knocking down the voltage-independent mAChR-A-KK should abolish its effects. We hypothesize that if the behavioral effect arising from mAChR-A-KK is localized to iLNs, overexpression of a *wt* mAChR-A or knockdown of mAChR-A-KK in KCs of the MB should not change mAChR-A-KK flies' behavior. As expected, applying both strategies to iLNs resulted in no odor habituation as was seen in *wt* flies (Fig. 5B, D), whereas, rescue attempts in KCs had no effect whatsoever on odor habituation and strong odor habituation was as observed as in the mAChR-A-KK fly strain. Together, these results support the notion that mAChR-A-KK induced potentiation of the ORN-iLN synapse underlies the increased

odor habituation observed. Taken together, our results unambiguously demonstrate that the voltage dependence of mAChR-A is crucial for both physiological process and behavioral output.

## Discussion

Here we show that the *Drosophila* mAChR-A is voltage dependent. In addition, our results reveal that a voltage-independent receptor variant, mAChR-A-KK, exhibits altered neuronal potentiation to both artificial and physiological stimuli, and that these changes in potentiation influence behavior. Furthermore, generating conditions such that mAChR-A becomes voltage independent and resides in the low activity state, completely abolishes the mAChR-A dependent potentiation. Thus, this work represents a demonstration that voltage dependence is crucial for the normal function of GPCRs in vivo, thereby changing our understanding of GPCR recruitment and function.

GPCRs voltage dependence was discovered almost 20 years ago[10]. Since then it was demonstrated for various GPCRs[7–15,56]. These studies provided important information on the identity of voltage-dependent GPCRs as well as on some of the mechanisms underlying GPCR voltage dependency. However, as they were performed in cell culture, whether this GPCR voltage dependency plays any physiological role was not addressed. GPRC voltage dependence was shown to control synaptic release initiation and duration in vitro[17–22], and recently it was shown that membrane depolarization recruits voltage-dependent purinergic receptors in sympathetic chromaffin cells to increase the quantal size[18]. Nevertheless, even in these cases, there is no evidence that these small changes in the duration or strength of synaptic release affect neuronal computation or behavioral output, especially on the background of noisy neural activity. Our study, which generated a fly strain with a voltage independent muscarinic receptor allowed us to address this question and unequivocally demonstrate that GPCR voltage dependence affects neuronal computation and behavioral output. Furthermore, contrary to all previous studies which showed effects only on synaptic release, this study demonstrates that GPCR voltage dependence plays a role post-synaptically in the canonical GPCR pathway.

Since mAChR-A voltage dependence shifts the dynamic range of the dose–response curve, it can only be relevant if mAChR-A is exposed to sub-saturation concentrations of neurotransmitter. In this context, mammalian glutamatergic receptors are usually far from saturation during quantal transmission[57]. Notably, the rapid removal of neurotransmitter from the synaptic cleft also generates sub-saturation conditions[58]. Our results further support this notion since mAChR-A-KK has similar activity to mAChR-A at saturation levels of the receptor (Fig. 1), and differs only in the dynamic range of responses. The findings of a strong physiological effect in response to synaptic release as well as a strong behavioral effect all point to a sub-saturating agonist concentration.

The demonstration that GPCR voltage dependence has physiological implications, suggests the presence of a strong crosstalk between the ionotropic pathways (that rapidly affect membrane potential) and the metabotropic pathways. For GPCRs that exhibit increased activity upon depolarization, this crosstalk is reminiscent of the crosstalk between the glutamatergic α-amino-3-hydroxy-5-methyl-4-isoxazolepropionic acid receptor (AMPAR) and N-methyl-D-aspartate receptor (NMDAR). NMDAR activity requires depolarization, which usually originates from AMPAR activation, and the recruitment of NMDAR can result in AMPA potentiation[59]. Although the metabotropic receptors do not require depolarization for their activity, depolarization favors their recruitment. For nicotinic and muscarinic receptors, a model arises where depolarization caused by

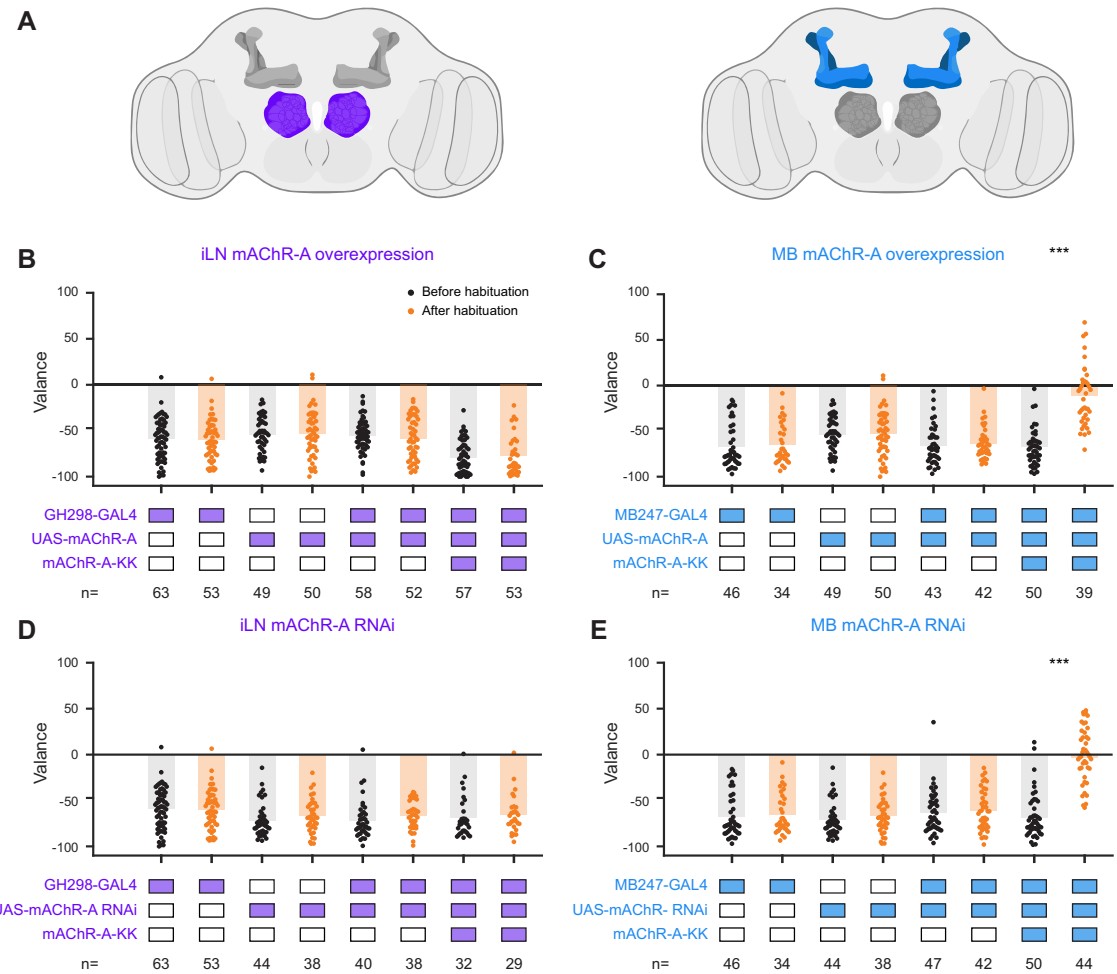

**Fig. 5 mAChR-A in iLNs is required for odor habituation. A** Schematics of the brain regions and neurons in the olfactory pathway that show high levels of mAChR-A expression[23, 28]. iLNs of the AL are labeled with purple and Kenyon cells (KCs) of the mushroom body (MB) are labeled with blue. **B–E** The valence (negative values indicate aversion) flies assign to ethyl butyrate before and after the odor habituation protocol (30 min exposure) were examined when the effect of the mAChR-A-KK mutation was abolished in iLNs (**B, D**, using the GH298-GAL driver line) or in KCs (**C, E**, using the MB247-GAL4 driver line). To abolish the effect of the mAChR-A-KK mutation we used either overexpression of *wt* mAChR-A (**B, C**) or knockdown of the mAChR-A-KK using RNAi (**D, E**). While reducing mAChR-A-KK in iLNs using both methods abolished the observed strong odor habituation, similar manipulation in KCs of the MB had no effect and flies still showed strong odor habituation ($n = 33–64$, p(panel **C**) $< 10^{-10}$, p(panel **E**) $< 10^{-10}$, Two-sample two sided *t*-test). GH298-GAL4 data are the same for panels **B** and **D**, MB247-GAL4 data are the same for panels **C** and **D**. Source data are provided as a Source Data file.

activation of nicotinic receptors strongly affects the activity of the co-activated muscarinic receptor. In turn, the improved recruitment of the muscarinic receptors results in potentiation of the nicotinic receptors (Fig. 3A). The above suggestion is not limited only to the muscarinic receptors but can also be extended to other GPCRs. For example, we have recently demonstrated a strong similarity between the function of mAChR-A in iLNs and the function of the glutamatergic mGluR1 expressed in inhibitory granule cells of the mammalian olfactory bulb[23,60]. It is interesting to note that this similarity is further extended to their voltage dependence and mGluR1 shifts to a high activity state following depolarization[14]. It is thus possible that similar to the *Drosophila* muscarinic receptor, mGluR1 voltage dependency participates in a crosstalk between glutamatergic ionotropic and metabotropic receptors and has a role in mammalian olfactory processing.

In *Drosophila*, odor habituation is thought to be mediated by the potentiation of iLNs[41,51]. Our results (Figs. 4 and 5) are in agreement with the role of iLNs in odor habituation and

demonstrate that cholinergic neuromodulation by mAChR-A is an important step in odor habituation. Our results further show that mAChR-A voltage dependence plays a role in odor habituation (Figs. 4 and 5). It is interesting to note that odor habituation requires PN input onto iLNs[54]. The requirement of PN input can explain how odor habituation occurs in a glomerulus-specific manner (i.e. according to the identity of the activated PNs)[51]. However, PNs generally respond in a non-linear manner to different concentrations, with relatively low odor concentrations required to saturate PN activity[61–63]. Thus, even low odor concentrations should generate strong odor habituation, contrary to observed results[55]. Our results, which show that mAChR-A voltage dependence plays a role in odor habituation (Figs. 4 and 5), may resolve this discrepancy. Contrary to PNs, iLNs are linearly recruited by ORNs[39]. Thus at low odor concentration iLNs will undergo only weak depolarization. Therefore, although the relevant PN will already be saturated, due to the low iLN depolarization, the responding mAChR-A will be in the low activity state and the overall habituation will not be strong. In contrast, at

a high odor concentration, iLNs will undergo strong depolarization, shifting mAChR-A to the high activity state, and as a result habituation will be stronger. In this way, GPCR voltage dependence can act as a rheostat that allows for a gradual increase in neuromodulation.

Taken together, this work provides a demonstration of a physiological role for the voltage dependency of GPCRs and may serve as a paradigm shift in our understanding of neural function and drug discovery.

## Methods

**Fly strains**. Fly strains (see below) were raised on cornmeal agar under a 12 h light/12 h dark cycle at 25 °C. The following transgenes were used: GH298-GAL4 (Bloomington #37294), MB247-GAL4[64], UAS-mAChR-A RNAi (TRiP.JF02725, Bloomington #27571), UAS-mCD8-GFP (Bloomington #32185), UAS-GCaMP6f (Bloomington #42747) and w[1118] (Bloomington #32185 5905).

The mAChR-A-KK fly strain was generated by GenetiVision (see supplemental methods). UAS-mAChR-A was inserted into the *Drosophila* pBID-UASC plasmid and flies were generated by BestGene Inc.

**Oocytes**. Xenopus laevis oocytes were isolated and incubated in NDE96 solution composed of ND96 (96 mM NaCl, 2 mM KCl, 1 mM CaCl$_2$, 1 mM MgCl$_2$, and 5 mM Hepes, pH adjusted to 7.5 with NaOH), with the addition of 2.5 mM Na$^+$ pyruvate, 100 U/ml penicillin and 100 mg/ml streptomycin[65]. One day after their isolation, the oocytes were injected with the following cRNAs: mAChR-A and its mutations (1 ng/oocyte) GIRK1 and GIRK2 (200 pg/oocyte for each) and Gαi3 (2 ng/oocyte). For experiments designed to measure β-arrestin activation cRNAs of GRK3 (1 ng/oocyte) and β-arrestin 2 (5 ng/oocyte) were co-injected. Gq activated Ca$^{2+}$ dependent Cl$^-$ currents were measured in oocytes injected with 5 ng/oocyte of the mAChR-A without the co-injection of other cRNAs.

cDNA plasmids were linearized with the appropriate restriction enzymes. Point mutations were prepared using Quick-Change II Site-Directed Mutagenesis Kit (Stratagene, La Jolla,CA, USA).

All experimental procedures used in this study were performed in accordance with relevant guidelines and regulations, and were approved by the Hebrew University's Animal Care and Use Committee (Ethical approval number NS-11-12909-3).

The currents were recorded 3–5 days after cRNA injection by using two electrode voltage clamp amplifier[10] (Warner OC 725 C amplifier, Warner Instruments, Hamden, CT). An oocyte was placed in the recording bath containing ND96 solution and was impaled by two electrodes pulled from 1.5 mm borosilicate capillaries (Warner instruments). Both electrodes were filled with 3 M KCl solution. The electrode resistance was between 0.5 and 2 MΩ. mAChR-A mediated GIRK currents were measured in a solution of 24 mM K$^+$, 72 mM NaCl, 24 mM KCl, 1 mM CaCl$_2$, 1 mM MgCl$_2$, and 5 mM Hepes, pH adjusted to 7.5 with KOH. mAChR-A mediated Ca$^{2+}$ dependent Cl$^-$ currents were measured in ND96 solution. pCLAMP10 software (Axon Instruments) was used for data acquisition.

The dose–response curves were fitted by the following equation: $Y = Bottom + X*(Top-Bottom)/(EC50 + X)$, where Y is the normalized response, X is the concentration of ACh, and EC50 is the ACh concentration that gives the half-maximal response. β-arrestin mediated desensitization was calculated as the percentage of GIRK current that remains after 80 sec continuous ACh application from the peak GIRK current.

**Olfactory stimulation**. Ethyl butyrate was purchased from Sigma-Aldrich (Rehovot, Israel) and was at the purest level available. ACV was bought at a local supermarket (RAUCH Fruchtsäfte GmbH & Co OG apple cider vinegar). Odors at $10^{-1}$ dilution were delivered by switching mass-flow controlled carrier at 0.4 l/min and stimulus stream (ms at 0.4 l/min (Sensirion) via software-controlled solenoid valves (The Lee Company). This resulted in a final dilution of $5 \times 10^{-2}$ of odor delivered to the fly. Air-streamed odor was delivered through a 1/16 inch ultra-chemical-resistant Versilon PVC tubing (Saint-Gobain, NJ, USA) that was placed 5 mm from the fly's antenna.

**Electrophysiology**. Flies were anesthetized on ice, then a single fly was moved to a custom-built chamber and fixed to aluminum foil using wax. The cuticle and trachea in the required area were removed, and the exposed brain was superfused with carbonated solution (95% O$_2$, 5% CO$_2$) containing 103 mM NaCl, 3 mM KCl, 5 mM trehalose, 10 mM glucose, 26 mM NaHCO$_3$, 1 mM NaH$_2$PO$_4$, 1.5 mM CaCl$_2$, 4 mM MgCl$_2$, and 5 mM N-Tris (TES), pH 7.3. Nominal external Ca$^{2+}$ was the same except that no Ca$^{2+}$ was added. Sample sizes were based on previous reports in the field. In vivo whole-cell current clamp recordings were made as previously described[66] on 2–4 day old male and female flies. Briefly, a driver line was used to drive GFP in target neurons and the fly brains were visualized with a Scientifica SliceScope Pro 1000 upright microscope with a 40x water immersion objective. Patch pipettes with a resistance of 9–12 MΩ were filled with a solution

containing: 140 mM potassium aspartate, 10 mM Hepes, 1 mM KCL, 4 mM MgATP, 0.5 mM Na$_3$GTP, and 1 mM EGTA. The pH of the solution was adjusted to 7.3 and osmolarity to 265 mOsm. Voltage was acquired using an Axon Instruments MultiClamp 700B in current-clamp mode, then was digitized at 50 kHz, and low-pass filtered at 1 kHz. A small constant hyperpolarizing current was applied to maintain a membrane potential of −60mV upon achieving a gigaseal and during the subsequent break-in.

**Functional imaging**. Functional Imaging was performed as previously described[28,67,68] using a two-photon laser-scanning microscopy (DF-Scope installed on an Olympus BX51WI microscope). Flies were prepared as described for the electrophysiology experiments. Fluorescence was excited by a Ti-Sapphire laser (Mai Tai HP DS, 100 fs pulses) centered at 910 nm, attenuated by a Pockels cell (Conoptics) and coupled to a galvo-resonant scanner. Excitation light was focused by a 20X, 1.0 NA objective (Olympus XLUMPLFLN20XW), and emitted photons were detected by GaAsP photomultiplier tubes (Hamamatsu Photonics, H10770PA-40SEL), whose currents were amplified (Hamamatsu HC-130-INV) and transferred to the imaging computer (MScan 2.3.01). All imaging experiments were acquired at 30 Hz.

**High-frequency stimulation application**. Antennal nerve stimulation was as described previously[63]. Briefly, a glass pipette was pulled and broken to create a narrow opening that allowed the antennal nerve to be sucked in. A constant current stimulator (Digitimer, DS3 Isolated Current Stimulator) was controlled by custom LabView script. Shock intensity was selected on the basis of the minimal stimulus needed to generate an excitatory postsynaptic current that was stable for the baseline period. All stimuli were in the range of 150–230 mA, and each lasted 50 μs. The high-frequency stimulation (HFS) protocol was as follows: 5 stimuli at 0.016 Hz were used as a baseline followed by HFS at 100 Hz for 1 sec at the selected holding potential, followed by 10 stimuli at 0.016 Hz. The peaks of the excitatory postsynaptic current (EPSC) were extracted using custom MATLAB scripts. In cases when the electrical stimulus elicited an action potential, the extracted value was the current at which the action potential was initiated as indicated by a sharp change in the current derivative.

**Pharmacology**. The following drugs were used: atropine (Sigma-Aldrich #A0132), nicotine (Sigma-Aldrich #N3876), muscarine (Sigma-Aldrich #M6532), acetylcholine (Sigma-Aldrich #A6625) and TTX (Alomone Labs #T-550). In all cases, stock solutions were prepared were diluted to the final concentration before experiments. Drugs were applied either by bath application or were injected directly to the AL using a pico-injector (Harvard Apparatus, PLI-100).

**Behavioral experiment**. For behavior experiments, 7–10 days old male and female fed flies were used. Experiments were performed in a custom-built, fully automated apparatus as previously described[28,69,70]. Briefly, single flies were placed in clear polycarbonate chambers (length 50 mm, width 5 mm, height 1.3 mm). For the ACV experiment, starved flies were used. Starved flies were placed in a vial containing water-soaked filter paper 24 h prior to the experiment. All flies were backcrossed except for *wt* and mAChR-A-KK strains that cannot be backcrossed. All experiments were performed during the flies' active hours. Air or odor streams were presented from each side and converged at a central choice zone. Mass flow controllers (CMOSens PerformanceLine, Sensirion) were used to control air flow. A carrier flow (2.7 l/min) was combined with an odor stream (0.3 l/min) obtained by circulating the air flow through vials filled with a liquid odorant resulting in a 3 l/min total flow that was split between the 20 chambers. As a result, the total flow in each half chamber was 0.15 l/min. Two identical odor delivery systems delivered odors independently to each half of the chamber. Ethyl butyrate and ACV were prepared daily at 10 fold dilution in mineral oil or water respectively which was further 10 fold diluted with the carrier air stream to yield a final 100 fold odor dilution.

The 20 chambers were stacked in two columns each containing 10 chambers and were backlit by 940 nm LEDs (Vishay TSAL6400). Images were obtained by a MAKO CMOS camera (Allied Vision Technologies) equipped with a Computer M0814-MP2 lens. The apparatus was operated in a temperature controlled incubator (Panasonic MIR 154) at 25 °C. A virtual instrument written in LabVIEW 7.1 (National Instruments) extracted fly position data from video images and controlled the delivery of odors. Data were analyzed in MATLAB 2018a (The MathWorks).

For odor habituation experiments, an odor valence protocol was performed which was composed of odor from the left side and airflow from the right side of the chamber for 2 min followed by 2 min of opposite sides (i.e. odor from the right side of the chamber). This was followed by an odor habituation protocol in which the odor was presented from both sides of the chamber for 30 min or 1 h. After the habituation period the odor valence protocol was repeated. The valance index was calculated as (preference for the left side when it contains odor) – (preference for the left side when it contains air).

**Quantification and statistical analysis**. All statistical testing and parameter extraction were done using custom MATLAB code (The MathWorks, Inc.), Prism

6 (GraphPad), or SPSS (IBM Corp.). Significance was defined as a p-value smaller than 0.05 and all statistical tests were two-sided. For all figures, error bars and shaded areas represent the standard error of the mean (SEM).

For presentation, bar plots with dots were generated using the UnivarScatter MATLAB ToolBox (https://www.mathworks.com/matlabcentral/fileexchange/54243-univarscatter) and the shadedErrorBar function (https://github.com/raacampbell/shadedErrorBar) for shaded errors on traces.

**Reporting summary**. Further information on research design is available in the Nature Research Reporting Summary linked to this article.

## Data availability

Source data are provided with this paper. The data used to generate the figures are available in the following GitHub page: https://github.com/ParnasLab. Source data are provided with this paper.

## Code availability

The code used to generate the figures is available in the following GitHub page: https://github.com/ParnasLab.

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

## Acknowledgements

The authors thank Dr. Robert J. Kittel and Dr. Moran Rubinstein for comments on the manuscript. We thank the Bloomington Stock Center and the Vienna Drosophila RNAi Center for fly strains. Molecular graphics and analyses were performed with UCSF Chimera, developed by the Resource for Biocomputing, Visualization, and Informatics at the University of California, San Francisco, with support from NIH P41-GM103311. This work was supported by the Israel Science Foundation (ISF 343/18, MP), the European Research Council (676844, MP), the Deutsche Forschungsgemeinschaft (project number 408264519 to MP) and the Open University of Israel (Internal research grant, YBC).

## Author contributions

E.R.: conceptualization, methodology, investigation, formal analysis, software, writing–review & editing, visualization. M.T.: investigation, formal analysis. Y.B.C.: investigation, formal analysis, writing–review & editing, supervision, funding acquisition. M.P.: Initiated the project, conceptualization, methodology, investigation, formal analysis, software, writing–original draft, writing–review & editing, visualization, supervision, funding acquisition.

## Competing interests

The authors declare no competing interests.
