## [Peer Review File · Nature Communications]

GPCR Voltage Dependence Controls Neuronal Plasticity and BehaviorREVIEWER COMMENTS

Reviewer #1 (Remarks to the Author):

Review NCOMMS-21-24890-T. Rozenfeld et al. „ GPCR voltage dependence controls neuronal plasticity and behavior”

The manuscript describes a study designed to investigate the physiological role of the voltage dependence of a muscarinic receptor using *Drosophila melanogaster* as a model system. The study is the first of its kind to investigate the physiological impact of the long known and well described voltage dependence of a GPCR. In order to study the *in vivo* impact of voltage dependence of a GPCR it is necessary to generate a voltage independent mutant of the receptor with otherwise unaltered ligand binding as well as G protein (and arrestin) coupling properties. The authors chose to study the M1R-analog mAChR-A of *Drosophila* based on the known voltage sensitivity of M1R and the physiological role of mAChR-A in olfaction in flies. Therefore, the authors created a voltage-insensitive mutant of mAChR-A by inserting two point mutations in the N-terminal part of the 3rd intracellular loop (previously identified for mammalian mAChR) and characterized its ability to activate GIRK currents in *Xenopus Oocytes*. Based on the lack of voltage sensitivity of this receptor mutant and its agonist sensitivity, which resembled that of wt mAChR-A at +40 mV (high active state, sensitive to low Atropine), the authors decided to use this receptor mutant for further *in vivo* studies in flies.

CRISPR - Cas mediated mutation of endogenous mAChR-A allowed the authors to generate a fly strain with a voltage-independent mAChR-A-KK. The authors tested the function of these receptors in GABAergic interneurons (iLN) located in a single glomeruli in the antennal lobe, in which stimulation of mAChR-A leads to a calcium influx. Comparing the sensitivity of muscarine to stimulate this current at different potentials, resulted in the finding, that muscarine was more potent at 0 mV compared to -80 mV, and that this voltage-sensitivity was lost in the mAChR-A-KK mutant strain as the sensitivity to muscarine remained high at -80 mV. Thereby the authors describe voltage dependence of mAChR-A for the first time in these neurons. Having generated a *in vivo* model for the test of voltage sensitivity of mAChR-A the authors went on to show that a high frequency stimulation protocol led to a voltage dependent plasticity in respect to measured mAChR dependent potentiation of postsynaptic EPSCs, which was voltage independent in the mAChR-A-KK strain. Furthermore, a voltage dependent potentiation of EPSPs induced by a double pulse of nicotine to induce depolarization was detectable only in wt flies but not in mAChR-A-KK or mAChR-A-KO flies. Using an odor habituation protocol, the authors found that only in the wt flies but not in the mAChR-A-KK mutant strain, the firing rate in the iLNs was decreased after habituation. Furthermore, the behavior in respect to avoidance of ethyl butyrate (odor) before and after a habituation was altered in mAChR-A-KK mutant flies as they showed less avoidance after the 30 min odor habituation protocol compared to the wt-strain.

General comments: This is a highly interesting study with the potential of a great impact in the field of GPCR physiology. The approach to generate a voltage insensitive GPCR mutant with otherwise unaltered signaling properties and to knock-in this mutant into organisms is in my eyes exactly the way to untangle the impact of voltage dependence of GPCRs. Furthermore it is of great value to use this approach in order to study the impact of voltage dependence of a GPCR on well characterized neuronal networks and even behavior. Of course this approach relies heavily on the “quality control” of the voltage independent properties of the receptor mutant. Here is my first critique: The authors - for very good reasons - use the IL3-Mutation instead of the tyrosine-lid mutations for their studies. As the mutated region is very close to - if not within - the contact side of muscarinic receptors with G proteins it is very important to ensure equal coupling of the receptor to Gq proteins. The authors only show Gi coupling in the heterologous *Xenopus Oocyte* system and not Gq coupling. Since the authors address synaptic plasticity, the unaltered GRK2 and Arrestin coupling properties of the mutant versus wt receptor need to be tested as well. Since for the interpretation of the whole study it is not sufficient to demonstrate that the mutant receptor does not show significant differences compared to the wt receptor, but instead, that indeed all accessible aspects of direct downstream signaling events are indistinguishable. Only then the authors can conclude that the observed differences in synaptic plasticity can be directly linked to the voltage dependence of muscarinic receptors.

The second general point that in my eyes need some attention is the link between the results of odor habituation in electrophysiological versus behavioral experiments. Both in the results section and in the discussion, I miss an explanation why a loss of synaptic plasticity in response to long term odor exposure in the mAChR-A-KK strain (no difference between pre and post exposure responses) is associated with an enhanced behavioral habituation in these flies.

The third general point concerns the discussion: It would benefit by staying far closer to the discussion of the presented results in the context of the literature instead of being very speculative.

Specific comments:

1. The strength of the presented data could be enhanced if the sample size would be adjusted to the variance of the response: For example in Figure 3 C and D as well as in Fig. 4 C and D it appears to me that given that close to half of the data points for wt- and mutant- receptor look identical, and only a fraction are really different, the sample size chosen is too small.

2. Similarly, the number of agonist concentrations used to demonstrate the agonist sensitivity of mAChR-A-KK in Figure 1 is very low. At least one intermediate agonist concentration would help to better evaluate small changes in agonist sensitivity.

3. I consider the analysis of the tyrosine lid mutations as a pre-study, which is not of such an importance for the reader that a whole paragraph is needed here. One sentence would be enough, also the supplemental info in this case is not needed.

4. In Figure 4A the potentiation for the mutant is not representative at all (as seen in B and C), but just an extreme example. Better pick a representative example

5. L 210: .."Compared to the significant decrease observed in the case of wt flies, these results indicate potentiation of iLNs in the case of the mAChR-A-KK fly strain." For me the data don't indicate potentiation, but rather insensitivity towards long term odorant exposure.

6. L 243....." Since GPCR voltage dependence shifts the dynamic range of the dose response curve, it can only be relevant if the GPCR is exposed to subsaturation concentrations of neurotransmitter". This statement may be ok for muscarinic receptors, however, since in several cases the efficacy of GPCRs is regulated by voltage far more prominently than the affinity, it is not applicable for generalization.

Reviewer #2 (Remarks to the Author):

In this manuscript, Rozenfeld and colleagues examine the functional consequences of manipulating the voltage dependence of an mAChR. They report an engineered variant of the *Drosophila* mAChR-A, which they call mAChR-A-KK, in which two point mutations abolish receptor voltage dependence but retain its current at full activation when expressed in oocytes. This modified receptor can be shifted to a low activity state using a competitive antagonist (atropine) allowing the authors to specifically examine the role of voltage dependence in the role of mAChRs. Using whole-cell recordings from *Drosophila* olfactory iLNs, which express mAChR-A, they show that post-synaptic depolarization increases post-tetanic potentiation driven by presynaptic ORN stimulation, and that this potentiation is blocked by mAChR-A knockdown. They further show that in flies carrying the mAChR-A-KK mutations, potentiation is observed regardless of post-synaptic voltage and that addition of atropine leads to no potentiation at any voltage. Given the voltage dependence of mAChR, the authors suggest that activation through nAChRs could lead to greater potentiation by constitutively activating mAChR-A. Consistent with this they show that a pre-pulse of nicotine increases the size of post-synaptic potentials in iLNs in wt but not mAChR-A-KK flies. Finally the authors ask whether the voltage dependence of mAChR has behavioral consequences. They examine iLN responses to an odorant (ethyl butyrate) after prolonged odor exposure (30

minutes) and show that wt but not KK iLNs show an average decrease in the odor response. They then show that odor avoidance is reduced in KK flies after habituation but not in wt flies. On this basis they conclude that the voltage dependence of mAChRs has behavioral consequences. Overall this is an interesting study that convincingly demonstrates that the voltage dependent of mAChR-A can be abolished with the KK mutations and contributes to post-tetanic potentiation in iLNs. However, the nicotine potentiation and behavioral aspects of the manuscript are less compelling. Overall the logic of the behavioral experiments is rather convoluted and not enough details are given about how the behavioral experiments were made. In addition, the authors should present further timecourse data for the nicotine experiments.

Major comments

1) The overall logic and details of the behavioral experiments are not clear. As I understand it, the reduced depression of iLNs in KK vs wt flies is interpreted as a "potentiation" which would then lead to greater odor habituation. However, the physiological data here do not really demonstrate a potentiation of iLN relative to their own baseline, therefore it is difficult to say that the behavioral habituation observed arises from this change in iLN activity. Further, no details about the behavioral experiments are provided in the text. What was the assay? How were flies treated before testing (age, feeding state, time of day, etc.)? Were the wt and KK flies backcrossed appropriately? Where else in the nervous system is mAChR-KK expressed and what effects might this have on behavior? Overall I think this behavioral experiment needs quite a bit of work to show convincingly that it arises from the change in iLN activity shown, and the manuscript might be better off leaving it out unless these issues can be addressed. Alternatively, it is possible that some combination of temporally patterned odor input or co-activation of multiple receptor channels could provide a more convincing demonstration of the in vivo role of mAChR voltage dependence.

2) In the nicotine potentiation experiment, timecourses of responses should be shown to make it clearer what is being compared.

Minor comments:

1) Example data shown in Fig. 4A do not appear to be representative based on the group data in B and C.

2) line 103 "comprises of" not grammatical

3) line 65-67: "PNs generally respond in a concentration-invariant manner." I think this is an overstatement. PNs show steeper activation curves than ORNs but are in no way "concentration invariant".

Response to reviewers, manuscript number NCOMMS-21-24890-T

General

We would like to thank the reviewers for their thoughtful comments, which have led to a significant improvement in the quality of our study.

Reviewer #1

1. *The manuscript describes a study designed to investigate the physiological role of the voltage dependence of a muscarinic receptor using drosophila melanogaster as a model system. The study is the first of its kind to investigate the physiological impact of the long known and well described voltage dependence of a GPCR. In order to study the in vivo impact of voltage dependence of a GPCR it is necessary to generate a voltage independent mutant of the receptor with otherwise unaltered ligand binding as well as G protein (and arrestin) coupling properties. The authors chose to study the M1R-analog mAChR-A of Drosophila based on the known voltage sensitivity of M1R and the physiological role of mAChR-A in olfaction in flies. Therefore, the authors created a voltage -insensitive mutant of mAChR-A by inserting two point mutations in the N-terminal part of the 3rd intracellular loop (previously identified for mammalian mAChR) and characterized its ability to activate GIRK currents in Xenopus Oocytes. Based on the lack of voltage sensitivity of this receptor mutant and it's agonist sensitivity, which resembled that of wt mAChR-A at +40 mV (high active state, sensitive to low Atropine), the authors decided to use this receptor mutant for further in vivo studies in flies.*

CRISPR - Cas mediated mutation of endogenous mAChR-A allowed the authors to generate a fly strain with a voltage-independent mAChR-A-KK. The authors tested the function of these receptors in GABAergic interneurons (iLN) located in a single glomeruli in the antennal lobe, in which stimulation of mAChR-A leads to a calcium influx. Comparing the sensitivity of muscarine to stimulate this current at different potentials, resulted in the finding, that muscarine was more potent at 0 mV compared to -80 mV, and that this voltage-sensitivity was lost in the mAChR-A-KK mutant strain as the sensitivity to muscarine remained high at -80 mV. Thereby the authors describe voltage dependence of mAChR-A for the first time in these neurons. Having generated a in vivo model for the test of voltage sensitivity of mAChR-A the authors went on to show that a high frequency stimulation protocol let to a voltage dependent plasticity in respect to measured mAChR dependent potentiation of postsynaptic EPSCs, which was voltage independent in the mAChR-A-KK strain. Furthermore, a voltage dependent potentiation of EPSPs induced by a double pulse of nicotine to induce depolarization was detectable only in wt flies but not in mAChR-A-KK or mAChR-A-KO flies. Using a odor habituation protocol, the authors found that only in the wt flies but not in the mAChR-A-KK mutant strain, the firing rate in the iLNs was decreased after habituation. Furthermore, the behavior in in respect to avoidance of ethyl butyrate (odor) before and after a habituation was altered in mAChR-A-KK mutant flies as they showed less avoidance after the 30 min odor habituation protocol compared to the wt-strain.

General comments: This is a highly interesting study with the potential of a great impact in the field of GPCR physiology. The approach to generate a voltage insensitive GPCR mutant with otherwise unaltered signaling properties and to knock-in this mutant into organisms is in my eyes exactly the way to untangle the impact of voltage dependence of GPCRs. Furthermore it is of great value to use this approach in order study the impact of voltage dependence of a GPCR on well characterized neuronal networks and even behavior. Of course this approach relies heavily on the “quality control” of the voltage independent properties of the receptor mutant. Here is my first critique: The authors - for very good reasons – use the IL3-Mutation instead of the tyrosine-lid mutations for their studies. As the mutated region is very close to - if not within - the contact side of muscarinic receptors with G proteins it is very important to ensure equal coupling of the receptor to Gq proteins. The authors only show Gi coupling in the heterologous Xenopus Oocyte system and not Gq coupling. Since the author address synaptic plasticity, the unaltered GRK2 and Arrestin coupling properties of the mutant versus wt receptor need to be tested as well. Since for the interpretation of the whole study it is not sufficient to demonstrate that the mutant receptor does not show significant differences compared to the wt receptor, but instead, that indeed all accessible aspects of direct downstream signaling events are indistinguishable. Only then the authors can conclude that the observed differences in synaptic plasticity can be directly linked to the voltage dependence of muscarinic receptors.

We thank the reviewer for this comment. We have added new Figure Extended data 2, which shows that the finding found using GIRK channels could also be observed for the G_q pathway as can be measured with Cl⁻ currents.

To examine the efficient recruitment of the G_q pathway we note the following: First, we have previously demonstrated that *in vivo* mAChR-A activates G_q (Rozenfeld et al. 2019, Cell Reports). Therefore, the *in vivo* dose response data also indicates that the G_q pathway is functional since there is no difference between the *wt* and *mAChR-A-KK* at the maximal concentration. Second, we observed no difference in the currents elicited by saturating level of ACh in the experiment described above which examined Gq elicited activity in oocytes (added new Figure Extended data 2). Third and most important, we performed *in vivo* 2-photon functional imaging to directly examine G_q activity *in vivo*. We expressed the genetically encoded Ca²⁺ indicator GCaMP6f in iLNs and examined the Ca²⁺ response in *wt* or *mAChR-A-KK* flies. Experiments were performed in nominally 0 Ca²⁺ external solution with TTX to measure only Ca²⁺ from interacellular sources. No difference was observed. This is now presented in new Figure Extended data 2.

We have also added experiments (new Figure Extended data 3) showing that there is no difference between the *wt* receptor and the KK receptor in terms of the effects of GRK3 and β-Arrestin.

2. The second general point that in my eyes need some attention is the link between the results of odor habituation in electrophysiological versus behavioral experiments. Both in the results section and in the discussion, I miss an explanation why a loss of synaptic plasticity in response to long term odor exposure in the mAChR-A-KK strain (no difference between pre and post exposure responses) is associated with an enhanced behavioral habituation in these flies.

We have added to new Figure 4 a scheme that explains the connection between potentiation of iLNs and odor habituation. This is now explained in lines 218-224. We also increased the number on neurons patched before and after prolonged odor exposure (n=31) and show that in *wt* flies the majority of iLNs show dramatic decreased activity whereas in KK flies more than a third of iLNs actually show potentiation (new Figure 4C).

3. The third general point concerns the discussion: It would benefit by staying far closer to the discussion of the presented results in the context of the literature instead of being very speculative.

We have modified the discussion as suggested by the reviewer.

*4. The strength of the presented data could be enhanced if the sample size would be adjusted to the variance of the response: For example in Figure 3 C and D as well as in Fig. 4 C and D it appears to me that given that close to half of the data points for *wt*- and mutant- receptor look identical, and only a fraction are really different, the sample size chosen is to small.*

We have increased the sample size in the above mentioned experiments.

5. Similarly, the number of agonist concentrations used to demonstrate the agonist sensitivity of mAChR-A-KK in Figure 1 is very low. At least one intermediate agonist concentration would help to better evaluate small changes in agonist sensitivity.

We have added two more concentration points.

6. I consider the analysis of the tyrosine lid mutations as a pre-study, which is not of such an importance for the reader that a whole paragraph is needed here. One sentence would be enough, also the supplemental info in this case is not needed.

We have shortened the relevant part of the text. Since we are not constrained by the number of supplemental figures and information, we chose to keep the supplemental information about these mutations.

7. In Figure 4A the potentiation for the mutant is not representative at all (as seen in B and C), but just an extreme example. Better pick a representative example

We have replaced the representative traces (new Figure 4B).

8. L 210: .."Compared to the significant decrease observed in the case of wt flies, these results indicate potentiation of iLNs in the case of the mAChR-A-KK fly strain." For me the data don't indicate potentiation, but rather insensitivity towards long term odorant exposure.

We increased the number of neurons patched before and after prolonged odor exposure (n=31) and show that in wt flies the majority of iLNs show dramatic decreased activity whereas in KK flies more than a third of iLNs actually show strong potentiation (new Figure 4C).

9. L 243....." Since GPCR voltage dependence shifts the dynamic range of the dose response curve, it can only be relevant if the GPCR is exposed to subsaturation concentrations of neurotransmitter". This statement may be ok for muscarinic receptors, however, since in several cases the efficacy of GPCRs is regulated by voltage far more prominently than the affinity, it is not applicable for generalization.

We revised this sentence to only address mAChR-A. Line 282.

Reviewer #2

*In this manuscript, Rozenfeld and colleagues examine the functional consequences of manipulating the voltage dependence of an mAChR. They report an engineered variant of the *Drosophila* mAChR-A, which they call mAChR-A-KK, in which two point mutations abolish receptor voltage dependence but retain its current at full activation when expressed in oocytes. This modified receptor can be shifted to a low activity state using a competitive antagonist (atropine) allowing the authors to specifically examine the role of voltage dependence in the role of mAChRs. Using whole-cell recordings from *Drosophila* olfactory iLNs, which express mAChR-A, they show that post-synaptic depolarization increases post-tetanic potentiation driven by presynaptic ORN stimulation, and that this potentiation is blocked by mAChR-A knockdown. They further show that in flies carrying the mAChR-A-KK mutations, potentiation is observed regardless of post-synaptic voltage and that addition of atropine leads to no potentiation at any voltage. Given the voltage dependence of mAChR, the authors suggest that activation through nAChRs could lead to greater potentiation by constitutively activating mAChR-A. Consistent with this they show that a pre-pulse of nicotine increases the size of post-synaptic potentials in iLNs in wt but not mAChR-A-KK flies. Finally the authors ask whether the voltage dependence of mAChR has behavioral consequences. They examine iLN responses to an odorant (ethyl butyrate) after prolonged odor exposure (30 minutes) and show that wt but not KK iLNs show an average decrease in the odor response. They then show that odor avoidance is reduced in KK flies after habituation but not in wt flies. On this basis they conclude that the voltage dependence of mAChRs has behavioral consequences. Overall this is an interesting study that convincingly demonstrates that the voltage dependent of mAChR-A can be abolished with the KK mutations and contributes to post-tetanic potentiation in iLNs. However, the nicotine potentiation and behavioral aspects of the manuscript are less compelling. Overall the logic of the behavioral experiments is rather convoluted and not enough details are given about how the behavioral experiments were made. In addition, the authors should present further timecourse data for the nicotine experiments.*

1. The overall logic and details of the behavioral experiments are not clear. As I understand it, the reduced depression of iLNs in KK vs wt flies is interpreted as a "potentiation" which would then lead to greater odor habituation. However, the physiological data here do not really demonstrate a potentiation of iLN relative to their own baseline, therefore it is difficult to say that the behavioral habituation observed arises from this change in iLN activity. Further, no details about the behavioral experiments are provided in the text. What was the assay? How were flies treated before testing (age, feeding state, time of day, etc.)? Were the wt and KK flies backcrossed appropriately? Where else in the nervous system is mAChR-KK expressed and what effects might this have on behavior? Overall I think this behavioral experiment needs quite a bit of work to show convincingly that it arises from the change in iLN activity shown, and the manuscript might be better off leaving it out unless these issues can be addressed. Alternatively, it is possible that some combination of temporally patterned odor input or co-activation of multiple receptor channels could provide a more convincing

demonstration of the in vivo role of mAChR voltage dependence.

We thank the reviewer for this comment. We have added to new Figure 4 a scheme that explains the connection between potentiation of iLNs and odor habituation. This is now explained in lines 218-224. We also increased the number on neurons patched before and after prolonged odor exposure (n = 31) and show that in *wt* flies the majority of iLNs show dramatic decreased activity whereas in KK flies more than a third of iLNs actually show strong potentiation (new Figure 4C).

We have previously shown that in the olfactory system mAChR-A is mainly expressed in iLNs and Kenyon cells (KCs) of the mushroom body (Bielopolsky et al. 2019, eLife; Rozenfeld et al. 2019, Cell Reports). We now better refer to this in lines 243-245. To pinpoint where the mAChR-A-KK mutations plays a role, we added new experiments in which we reduce the effect of mAChR-A-KK in specific neurons. To this end, we replaced mAChR-A-KK by overexpressing *wt* mAChR-A or reduced its level by expressing mAChR-A RNAi on the background of KK flies in iLNs or KCs. We show that reducing the effect of mAChR-A-KK only in iLNs rescues odor habituation, whereas reducing the effect of mAChR-A-KK only in KCs has no effect on odor habituation (new Figure 5).

All details about the behavior experiments are now provided in the methods section and we refer to them in the main text (lines 225 and 743-771).

2. In the nicotine potentiation experiment, timecourses of responses should be shown to make it clearer what is being compared.

We have added to new Figure 3 traces of response to show what is being compared.

3. Example data shown in Fig. 4A do not appear to be representative based on the group data in B and C.

We have replaced the representative traces (new Figure 4B).

4. line 103 "comprises of" not grammatical

We fixed this mistake.

5. *line 65-67: "PNs generally respond in a concentration-invariant manner." I think this is an overstatement. PNs show steeper activation curves than ORNs but are in no way "concentration invariant".*

We fixed this mistake. The new text now reads, "PNs generally respond in a non-linear manner to different concentrations, with relatively low odor concentrations required to saturate PN activity"

REVIEWERS' COMMENTS

Reviewer #1 (Remarks to the Author):

The authors did a good job in addressing my comments and added results from important control experiments. I would love to see concentration response curves for the GRK- ARR - MR interaction experiments, since small differences between mutant and wt receptor would pop up more easily. However, showing kinetics under steady state conditions is also conclusive. Therefore I would recommend instead of comparing overall desensitization in the presence of ARR and GRK3 to show averaged traces normalized to the initial ACh response (extended Figure 3 A,B). This would allow to assess potential differences in desensitization kinetics, which in my opinion will be a more sensitive parameter than endpoint measurements under saturation conditions.

Overall the improvements of the manuscript made it more conclusive.

Reviewer #2 (Remarks to the Author):

The authors have addressed the majority of my concerns. The additional data supporting localization of the behavioral phenotype to iLNs are compelling.

However, I do think some brief description of the behavior needs to be included in the main results and not just referred to the methods. This could potentially just be a small diagram of the behavioral apparatus and a brief description of the habituation and valence protocol. It might also be helpful to include in the diagrams for Fig 5 something like the diagram in 4A to illustrate the expected outcomes for each experiment, and to slightly expand the text in lines 248-251 to spell out the logic and interpretation of the rescue experiments.

There are also a few typos in the text:

line 255: habitation

Response to reviewers, manuscript number NCOMMS-21-24890A

Reviewer #1 (Remarks to the Author):

The authors did a good job in addressing my comments and added results from important control experiments. I would love to see concentration response curves for the GRK- ARR - MR interaction experiments, since small differences between mutant and wt receptor would pop up more easily. However, showing kinetics under steady state conditions is also conclusive. Therefore I would recommend instead of comparing overall desensitization in the presence of ARR and GRK3 to show averaged traces normalized to the initial ACh response (extended Figure 3 A,B). This would allow to assess potential differences in desensitization kinetics, which in my opinion will be a more sensitive parameter than endpoint measurements under saturation conditions.

Overall the improvements of the manuscript made it more conclusive.

We have added to Supplemental Figure 3 a new panel with traces demonstrating the decay kinetics under steady state as requested.

Reviewer #2 (Remarks to the Author):

The authors have addressed the majority of my concerns. The additional data supporting localization of the behavioral phenotype to iLNs are compelling.

However, I do think some brief description of the behavior needs to be included in the main results and not just referred to the methods. This could potentially just be a small diagram of the behavioral apparatus and a brief description of the habituation and valence protocol. It might also be helpful to include in the diagrams for Fig 5 something like the diagram in 4A to illustrate the expected outcomes for each experiment, and to slightly expand the text in lines 248-251 to spell out the logic and interpretation of the rescue experiments.

Due to word limit in the figure caption, we have added a new supplemental Figure 6 with a diagram of the behavior chamber as well as examples of flies' walking trajectories in the behavior chambers. We have also added to the text a more detailed explanation. The text now reads "To test odor habituation we used custom linear chambers, each housing a single fly (Supplementary Figure 6A). These chambers allow presentation of an odor from either sides of the chamber while presenting odorless air flow from the other side of the chamber. The Air and odor streams converge at a central choice zone (see methods for a detailed description). Thus, each fly can choose whether to spend time in the area

containing an odor. To calculate ethyl butyrate valence the odor was presented on alternating sides of the chamber for two minutes and the difference between the preference to the odor containing side relative to the air containing side was calculated (Supplementary Figure 6B, see methods for a detailed explanation). Following an initial odor valence measurement, flies were exposed to ethyl butyrate for 30 minutes on both sides of the chamber simultaneously and then odor valence was measured again (Supplementary Figure 6B)".

In addition, we have better explained the rationale behind the RNAi and over expression experiments. The text now reads "Therefore, we performed rescue experiments in these two cell types. To this end we used the GH298-GAL4 driver which covers iLNs, and the MB247-GAL driver which covers the KC subtypes that express mAChR-A^{23,28}. To rescue the effects of mAChR-A-KK we used two strategies: i. we overexpressed a *wt* mAChR-A using UAS-mAChR-A. ii. We knocked down mAChR-A-KK levels using UAS-mAChR-A RNAi^{23,28}. Overexpression of a *wt* mAChR-A in iLNs should abolish the increased habituation effect in mAChR-A-KK flies since the overexpression of a voltage dependent mAChR-A has a dominant effect. Similarly, knocking down the voltage independent mAChR-A-KK should abolish its effects. We hypothesize that if the behavioral effect arising from mAChR-A-KK is localized to iLNs, overexpression of a *wt* mAChR-A or knockdown of mAChR-A-KK in KCs of the MB should not change mAChR-A-KK flies' behavior".

There are also a few typos in the text:

line 255: habitation

Thank you.